# Comparison of physician-delivered models of virtual and home-based in-person care for adults in the last 90 days of life with cancer and terminal noncancer illness during the COVID-19 pandemic

Rabia Akhter[1], Thérèse A. Stukel[2,3], Hannah Chung[2], Chaim M. Bell[1,3,4], Allan S. Detsky[1,3,4], James Downar[5,6,7], Sarina R. Isenberg[5,6,8], John Lapp[1,9], Media Mokhtarnia[1], Nathan Stall[1,4,10], Peter Tanuseputro[2,3,5,6,11], Kieran L. Quinn[1,2,3,4,12] *

1 Department of Medicine, Sinai Health System, Toronto, ON, Canada, 2 ICES, Toronto and Ottawa, ON, Canada, 3 Insitute of Health Policy, Management and Evaluation, University of Toronto, Toronto, ON, Canada, 4 Department of Medicine, University of Toronto, Toronto, ON, Canada, 5 Division of Palliative Care, Dept of Medicine, University of Ottawa, Ottawa, ON, Canada, 6 Bruyere Research Institute, Ottawa, ON, Canada, 7 Queensland University of Technology School of Law, Queensland, Australia, 8 Department of Family and Community Medicine, University of Toronto, Toronto, ON, Canada, 9 Northern Ontario School of Medicine University, Sudbury, ON, Canada, 10 Women's Age Lab, Women's College Hospital, Toronto, ON, Canada, 11 Ottawa Hospital Research Institute, Ottawa, ON, Canada, 12 Temmy Latner Centre for Palliative Care, Sinai Health System, Toronto, ON, Canada

* kieran.quinn@sinaihealth.ca

## Abstract

### Objective

To measure the association between types of serious illness and the use of different physician-delivered care models near the EOL during the COVID-19 pandemic.

### Design, setting and participants

Population-based cohort study using health administrative datasets in Ontario, Canada, for adults aged ≥18 years in their last 90 days of life who died of cancer or terminal noncancer illness and received physician-delivered care models near the end-of-life between March 14, 2020 and January 24, 2022.

### Exposure

The type of serious illness (cancer or terminal noncancer illness).

### Main outcome

Physician-delivered care models for adults in the last 90 days of life (exclusively virtual, exclusively home-based in-person, or mixed).

**Data Availability Statement:** The dataset from this study is held securely in coded form at ICES. While data-sharing agreements prohibit ICES from making the dataset publicly available, access may be granted to those who meet pre-specified criteria for confidential access, available at www.ices.on.ca/DAS. The full dataset creation plan and underlying analytic code are available from the authors upon request, understanding that the computer programs may rely upon coding templates or macros that are unique to ICES and are, therefore, either inaccessible or may require modification.

**Funding:** This study received funding from the Canadian Institutes of Health Research (CIHR PNN-177923), and Health Canada's Health Care Policy and Strategies Program. This study was also supported by the Innovation Fund of the Alternative Funding Plan for the Academic Health Sciences Centres of Ontario as well as ICES, which is funded by an annual grant from the Ontario Ministry of Health and the Ministry of Long-Term Care. This document used data adapted from the Statistics Canada Postal Code OM Conversion File, which is based on data licensed from Canada Post Corporation, and/or data adapted from the Ontario Ministry of Health Postal Code Conversion File, which contains data copied under license from ©Canada Post Corporation and Statistics Canada. Parts of this material are based on data and/or information compiled and provided by the Ontario Ministry of Health and the Canadian Institute for Health Information. The analyses, conclusions, opinions and statements expressed herein are solely those of the authors and do not reflect those of the funding or data sources; no endorsement is intended or should be inferred.

**Competing interests:** The authors have declared that no competing interests exist.

## Results

The study included 75,930 adults (median age 78 years, 49% female, cancer n = 58,894 [78%], noncancer illness n = 17,036 [22%]). A higher proportion of people with cancer (39.3%) received mixed model of care compared to those with noncancer illnesses (chronic organ failure 24.4%, dementia 37.9%, multimorbidity 28%). Compared to people with cancer, people with chronic organ failure (adjusted odds ratio [aOR], 1.61, 95% CI: 1.54 to 1.68) and those with multimorbidity ([aOR], 1.49, 95% CI: 1.39 to 1.59) had a higher odds of receiving virtual care than a mixed model of care. People with dementia had a higher odds of home-based in-person care than a mixed model of care ([aOR], 1.47, 95% CI 1.27, 1.71) and virtual care ([aOR], 1.40, 95% CI 1.20–1.62) compared to people with cancer.

## Conclusion

A person's type of serious illness was associated with different care models near the end-of-life. This study demonstrates persistent disease-specific differences in care delivery or possibly the tailoring of models of care in the last 90 days of life based on a person's specific care needs.

## Introduction

End-of-life (EOL) care is an essential component of healthcare that often addresses a period of burdensome symptoms, reduced quality of life, and focuses on providing comfort and support to people and their families during the final stages of a serious illness [1]. Despite being a critical aspect of healthcare, previous research identified differences in models of care near the EOL that were related to whether a person has cancer or terminal noncancer illness [2, 3]. Much of the prior focus on improving care near the EOL has been on people with cancer [4, 5].

The COVID-19 pandemic necessitated a significant shift towards virtual care, including physician-delivered care models near the EOL [6]. Virtual care utilizes video conferencing and telehealth to support people remotely and has potential benefits, including increased accessibility to care, improved communication between people and care teams, and the ability to provide care in the person's home [7–12]. On March 14, 2020, the Ontario Government introduced a set of reimbursable telephone and video-based provider fee codes to enable the delivery of virtual care, including in the last 90 days of life. During the pandemic, use of virtual care was utilized to reduce the risk of SARS-CoV-2 transmission from in-person visits. However, virtual care as an emerging health technology presents potential challenges that may limit its use and access among older adults, those with comorbidities, frailty, impaired cognitive or physical function, and pre-existing disabilities [13–15]. Therefore, utilizing a physician-delivered mixed model of in-person and virtual care may be optimal near the EOL. Indeed, a recent qualitative study examined the experiences and perspectives of healthcare professionals (HCPs), unpaid family caregivers, and people who delivered or received home-based in-person virtual care during the pandemic. It found that participants generally preferred in-person care to virtual care, but a mixed model of care delivery may be ideal for the future [16].

Despite observed differences in EOL care delivery models between people with different types of serious illness before the pandemic, it remains unknown if these differences persisted in the pandemic during a period of widespread virtual care use. Further, it is unknown if a

preferential model of care is used more frequently among people with cancer compared to people with terminal noncancer illness.

To address these existing knowledge gaps in patterns of EOL care delivery during the pandemic, we measured the association between types of serious illness (cancer and noncancer terminal illness: chronic organ failure, dementia, multimorbidity) and the use of different physician-delivered care models (exclusively virtual, exclusively home-based in-person, and mixed model of care) near the EOL.

## Methods

This study is reported in compliance with guidelines for The Reporting of studies Conducted using Observational Routinely collected health Data (RECORD) (**S1 Table**) [17].

### Ethics

ICES is a prescribed entity under Ontario's Personal Health Information Protection Act (PHIPA). Section 45 of the Personal Health Information Protection Act (PHIPA) permits ICES to collect and use personal health information (PHI) without consent of the individual to whom the PHI relates if the collection/use is for the purpose of Evaluation, Planning or Management (EPM) of all or part of the health system and/or health services. ICES does not require Research Ethics Board (REB) approval for ICES Projects that involve use of PHI and are conducted entirely for EPM purposes.

### Study design, setting and data sources

We performed a retrospective population-based cohort study using linked clinical and administrative datasets in Ontario, Canada. These datasets were linked using unique encoded identifiers and analyzed at ICES (formerly the Institute for Clinical and Evaluative Sciences) and are commonly used for studies including models of care for adults in the last 90 days of life [3, 18–22]. The use of population-level linked administrative data minimizes selection bias and improves the diversity of the study cohort to strengthen the overall generalizability of the study and its principal findings.

Ontario is Canada's most populous province, with over 15 million adults. Residents of Ontario have public insurance for hospital and home care and physicians' services, and those aged $\geq$ 65 years are provided prescription drug insurance.

### Study cohort

We included all Ontario adults aged $\geq$18 years who died with cancer or terminal noncancer illness and received physician-delivered models of care in the 90 days before death between March 14, 2020 and January 24, 2022. The index date was 90 days before the person's date of death.

We excluded adults 1) whose last 90 days of life started before March 14, 2020, which preceded the onset of the pandemic and the majority of resulting health system changes; 2) without prevalent cancer, chronic organ failure, or dementia at 90 days before death (the primary exposures); 3) residents living in nursing homes in the past two years as care delivery during the pandemic was significantly interrupted in this setting; and 4) with $\leq$1 visit or who were institutionalized during the last 90 days of life. The need to have 2 or more visits was necessary to determine if people received exclusively in-person, exclusively virtual, or mixed models of physician-delivered care in the last 90 days of life (the primary outcome). We also excluded individuals who were non-Ontario residents, ineligible for medical care through the Ontario

Health Insurance Plan for more than 90 days continuously in the prior year and therefore could not receive publicly insured health services, and people who did not access the Ontario healthcare system at least once in the past 10 years to ensure people are still residing in the province.

## Exposure

The primary exposure was a person's type of serious illness, which was classified as cancer or terminal noncancer illness. We determined the type of prevalent serious illness at the index date based on validated ICES algorithms. Terminal noncancer illnesses were defined as chronic organ failure (COPD, heart failure, stroke, diabetes, end-stage renal disease, hypertension & severe liver disease) and dementia because these diseases are the most prevalent terminal noncancer illness and are also associated with high healthcare utilization and reduced quality of life [2, 3, 19, 23–27]. Multimorbidity was defined as having ≥3 prevalent chronic terminal noncancer conditions. For the purpose of the analyses, individuals with cancer were assigned to a mutually exclusive group only and not included in the other exposure groups. Individuals with more than 1 noncancer illness could appear in more than one group (e.g., in both the chronic organ failure and dementia groups). We intentionally did not include people with cancer in the multimorbidity group because our primary study objective was to compare care delivery between these individuals to those with terminal noncancer illnesses. People with cancer often receive care through a specialized center that is highly resourced and, therefore, more likely to deliver care differently.

## Characteristics of the study cohort

We measured a person's demographic and clinical variables consisting of age, sex, surname-based ethnicity [28], neighbourhood socioeconomic status, rural residence, chronic health conditions [29], and hospital frailty risk score, using a five-year look-back period from the index date.

 We measured the characteristics of physicians who provided care to patients in their last 90 days of life in each serious illness group, including age, sex, education, physician practice location, number of years in practice, specialty, status as a palliative care specialist, and the number and type of visits in the calendar year of index.

## Outcomes

The primary outcome was the use of three distinct models of physician-delivered care in the last 90 days of life: 1) exclusively virtual, 2) exclusively home-based in-person, and 3) mixed virtual and home-based in-person care. These models were determined using a distinct set of physician fee codes specifying the type and location of care delivery (**S2 Table**). In Ontario, the majority of EOL care is provided by physicians in collaboration with multidisciplinary teams. The secondary outcome was the number of visits across each type of care model delivered before death for each of the different serious illness groups. These were measured using the physician (i.e., number of EOL visits per patient per physician) as the unit of analysis to reflect the average intensity of care required by each physician across each care model.

## Statistical analysis

Multivariable, multinomial logistic regression was used to measure the association between the type of serious illness and the use of exclusively virtual or home-based in-person care, and mixed models of care in the last 90 days of life. The main comparison of interest was the

receipt of exclusively virtual versus mixed models of care because it is believed virtual care is best delivered in this way [16, 30]. In all analyses, cancer was used as the referent group. Models were adjusted for all baseline patient characteristics except the prevalent chronic conditions under study. All analyses were performed using SAS version 9.4 (SAS Institute, Cary, North Carolina).

## Results

### Baseline characteristics

There were 75,930 adults whose last 90 days of life was between March 14, 2020 and January 24, 2022 included in the study cohort. Among them, 58,894 (78%) had cancer and 17,036 (22%) had terminal noncancer illness (16,096 had chronic organ failure, 3,908 had dementia, and 5,246 had multimorbidity–groups are not mutually exclusive) at the time of their death (Fig 1).

Compared to people with cancer, people with chronic organ failure were of similar age and sex and equally likely to reside in rural areas. People with multimorbidity were more likely to be of South Asian ethnicity, lower socioeconomic status, and have heart failure, COPD, or diabetes. People with dementia tended to be older, female, and more likely to reside in an urban area (Table 1).

The physicians providing care to the study cohort shared similar characteristics with each other, such as age, sex, education, practice location and years in practice. A higher proportion of physicians who provided care to people with dementia were general practitioners and palliative care specialists compared to those who provided care to people with cancer (S3 Table).

### Association of the type of serious illness with models of care in the last 90 days of life

Overall, the majority of people received exclusively virtual care in the last 90 days of life regardless of their type of serious illness compared to other models of care (Table 2, S4 Table). A higher proportion of people who died with chronic organ failure (72.5%) and multimorbidity (68.9%) received exclusively virtual care in the last 90 days of life compared to those who died with cancer (57.4%). A lower proportion of people who died with dementia (55.6%) received exclusively virtual care in the last 90 days of life compared to those who died with cancer. A higher proportion of people with cancer (39.3%) received mixed-model care in the last 90 days of life compared to those with noncancer illness (Chronic organ failure 24.4%, dementia 37.9% and multimorbidity 28%) (Table 2, Fig 2).

Compared to people with cancer, people with chronic organ failure had higher odds of receiving exclusively virtual care than mixed care in the last 90 days of life (adjusted odds ratio [aOR], 1.61,95% CI: 1.54 to 1.68) as did people with multimorbidity ([aOR], 1.49, 95% CI: 1.39 to 1.59). The odds of receiving exclusively virtual care and mixed model care in the last 90 days of life were similar among people with dementia (aOR 1.05, 95% CI: 0.98 to 1.13), but the odds of exclusively home-based in-person care were higher than a mixed model care ([aOR], 1.47, 95% CI 1.27, 1.71) or exclusively virtual care ([aOR], 1.40, 95% CI 1.20–1.62) than among people with cancer (Table 2, Fig 2, S4 Table).

On average, physicians delivered a similar number of visits near the EOL per patient across each type of serious illness (mean ± SD: cancer, 2.1 ± 1.5; chronic organ failure, 2.1 ± 1.4; dementia, 2.4 ± 1.5; multimorbidity, 2.1 ± 1.5).

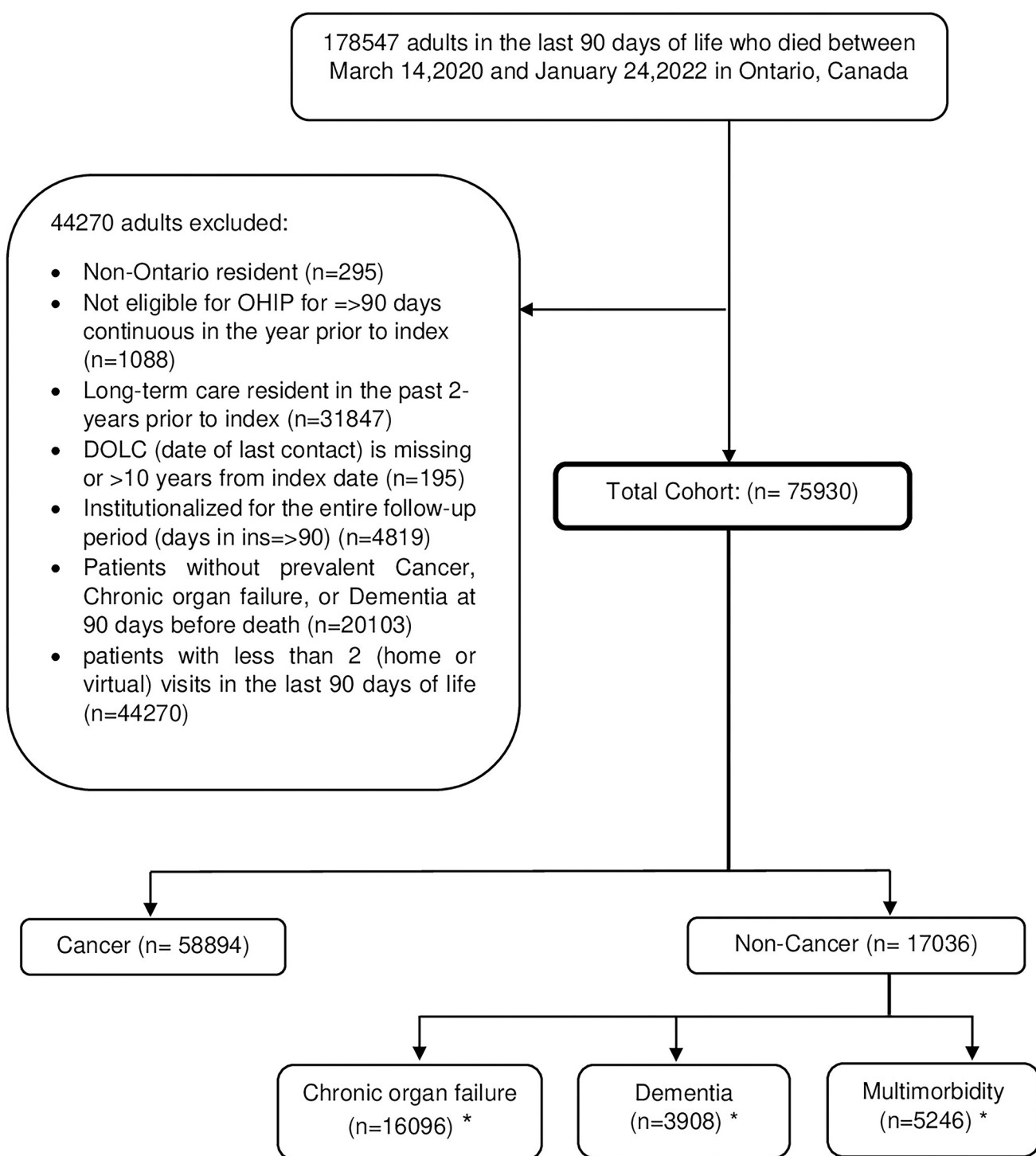

*Groups are not mutually exclusive (e.g., a dementia patient could also be in the chronic organ failure group).

**Fig 1. Creation of the study cohort.**

**Table 1. Baseline characteristics of adults in their last 90 days of life according to type of serious illness.**

| | Cancer N = 58894 | Chronic Organ Failure N = 16096 | Dementia N = 3908 | Multimorbidity N = 5246 | Standardized differences | | |
| --- | --- | --- | --- | --- | --- | --- | --- |
| | | | | | Cancer vs Chronic organ failure | Cancer vs Dementia | Cancer vs multimorbidity |
| **Age** | | | | | | | |
| **Median (IQR)** | 78 (69–87) | 79 (69–87) | 86 (79–91) | 81 (73–88) | 0.02 | 0.69 | 0.23 |
| **18–29** | 157 (0.3%) | 54 (0.3%) | 0 (0.0%) * | < = 5 (0.1%) * | 0.01 | 0.07 | 0.04–0.06* |
| **30–39** | 439 (0.7%) | 124 (0.8%) | 0 (0.0%) * | < = 5 (0.1%) * | 0 | 0.12 | 0.10–0.12* |
| **40–49** | 1,213 (2.1%) | 365 (2.3%) | 11 (0.3%) | 50 (1.0%) | 0.01 | 0.17 | 0.09 |
| **50–59** | 4,148 (7.0%) | 1,135 (7.1%) | 45 (1.2%) | 209 (4.0%) | 0 | 0.3 | 0.13 |
| **60–69** | 9,829 (16.7%) | 2,579 (16.0%) | 192 (4.9%) | 640 (12.2%) | 0.02 | 0.39 | 0.13 |
| **70–79** | 15,669 (26.6%) | 4,150 (25.8%) | 750 (19.2%) | 1,376 (26.2%) | 0.02 | 0.18 | 0.01 |
| **80–89** | 17,784 (30.2%) | 4,959 (30.8%) | 1,669 (42.7%) | 1,974 (37.6%) | 0.01 | 0.26 | 0.16 |
| **90+** | 9,655 (16.4%) | 2,730 (17.0%) | 1,241 (31.8%) | 989 (18.9%) | 0.02 | 0.37 | 0.06 |
| **Sex** | | | | | | | |
| **Female** | 28,330 (48.1%) | 7,073 (43.9%) | 2,123 (54.3%) | 2,296 (43.8%) | 0.08 | 0.12 | 0.09 |
| **Male** | 30,564 (51.9%) | 9,023 (56.1%) | 1,785 (45.7%) | 2,950 (56.2%) | 0.08 | 0.12 | 0.09 |
| **Neighbourhood income quintile** | | | | | | | |
| **1 (lowest)** | 12,680 (21.5%) | 4,272 (26.5%) | 867 (22.2%) | 1,377 (26.2%) | 0.12 | 0.02 | 0.11 |
| **2** | 12,615 (21.4%) | 3,740 (23.2%) | 907 (23.2%) | 1,286 (24.5%) | 0.04 | 0.04 | 0.07 |
| **3** | 11,837 (20.1%) | 3,244 (20.2%) | 820 (21.0%) | 1,026 (19.6%) | 0 | 0.02 | 0.01 |
| **4** | 10,577 (18.0%) | 2,622 (16.3%) | 650 (16.6%) | 869 (16.6%) | 0.04 | 0.04 | 0.04 |
| **5 (highest)** | 10,969 (18.6%) | 2,145 (13.3%) | 639 (16.4%) | 664 (12.7%) | 0.15 | 0.06 | 0.16 |
| **Missing** | 216 (0.4%) | 73 (0.5%) | 25 (0.6%) | 24 (0.5%) | 0.01 | 0.04 | 0.01 |
| **Rural residence** | | | | | | | |
| **No** | 52,167 (88.6%) | 14,384 (89.4%) | 3,602 (92.2%) | 4,719 (90.0%) | 0.03 | 0.12 | 0.04 |
| **Yes** | 6,529 (11.1%) | 1,649 (10.2%) | 283 (7.2%) | 505 (9.6%) | 0.03 | 0.13 | 0.05 |
| **Missing** | 198 (0.3%) | 63 (0.4%) | 23 (0.6%) | 22 (0.4%) | 0.01 | 0.04 | 0.01 |
| **Ethnicity, n (%)** | | | | | | | |
| **Chinese** | 1,614 (2.7%) | 626 (3.9%) | 181 (4.6%) | 179 (3.4%) | 0.06 | 0.1 | 0.04 |
| **General** | 56,244 (95.5%) | 14,561 (90.5%) | 3,494 (89.4%) | 4,750 (90.5%) | 0.2 | 0.23 | 0.2 |
| **South Asian** | 1,030 (1.7%) | 905 (5.6%) | 231 (5.9%) | 316 (6.0%) | 0.21 | 0.22 | 0.22 |
| **Missing** | 6 (0.0%) | < = 5 (0.0%) | < = 5 (0.1%) | < = 5 (0.0%) | 0.01 | 0.02 | 0.01 |
| **Chronic Disease** | | | | | | | |
| **Heart failure** | 16,154 (27.4%) | 6,981 (43.4%) | 1,316 (33.7%) | 4,111 (78.4%) | 0.34 | 0.14 | 1.19 |
| **COPD** | 12,517 (21.3%) | 4,568 (28.4%) | 679 (17.4%) | 2,262 (43.1%) | 0.17 | 0.1 | 0.48 |
| **Dementia** | 8,378 (14.2%) | 2,968 (18.4%) | 3,908 (100.0%) | 1,777 (33.9%) | 0.11 | 3.47 | 0.47 |

*(Continued)*

**Table 1.** (Continued)

| | Cancer N = 58894 | Chronic Organ Failure N = 16096 | Dementia N = 3908 | Multimorbidity N = 5246 | Standardized differences | | |
| --- | --- | --- | --- | --- | --- | --- | --- |
| | | | | | Cancer vs Chronic organ failure | Cancer vs Dementia | Cancer vs multimorbidity |
| **Severe liver disease** | 1,015 (1.7%) | 343 (2.1%) | 18 (0.5%) | 145 (2.8%) | 0.03 | 0.12 | 0.07 |
| **Diabetes** | | 10,129 (62.9%) | 1,837 (47.0%) | 4,321 (82.4%) | 0.51 | 0.18 | 1.02 |
| **Hypertension** | 44,970 (76.4%) | 13,587 (84.4%) | 3,301 (84.5%) | 5,001 (95.3%) | 0.2 | 0.21 | 0.57 |
| **End-stage renal disease** | 15,881 (27.0%) | 6,366 (39.6%) | 1,297 (33.2%) | 3,948 (75.3%) | 0.27 | 0.14 | 1.1 |
| **Stroke** | 6,295 (10.7%) | 2,571 (16.0%) | 753 (19.3%) | 1,469 (28.0%) | 0.16 | 0.24 | 0.45 |
| **Psychotic disorder** | 631 (1.1%) | 345 (2.1%) | 108 (2.8%) | 87 (1.7%) | 0.09 | 0.12 | 0.05 |
| **Non-psychotic disorder** | 15,142 (25.7%) | 3,824 (23.8%) | 1,065 (27.3%) | 1,227 (23.4%) | 0.05 | 0.03 | 0.05 |
| **Alcohol and substance use disorder** | 1,629 (2.8%) | 827 (5.1%) | 75 (1.9%) | 148 (2.8%) | 0.12 | 0.06 | 0 |
| **Hospital frailty risk score** | | | | | | | |
| **0. 0** | 8,376 (14.2%) | 1,186 (7.4%) | 116 (3.0%) | 202 (3.9%) | 0.22 | 0.41 | 0.37 |
| **0.1–4.9** | 14,905 (25.3%) | 3,266 (20.3%) | 515 (13.2%) | 941 (17.9%) | 0.12 | 0.31 | 0.18 |
| **5.0–8.9** | 7,794 (13.2%) | 2,224 (13.8%) | 499 (12.8%) | 967 (18.4%) | 0.02 | 0.01 | 0.14 |
| **9.0 +** | 12,087 (20.5%) | 4,378 (27.2%) | 1,638 (41.9%) | 2,567 (48.9%) | 0.16 | 0.47 | 0.63 |
| **No prior hospitalizations** | 15,732 (26.7%) | 5,042 (31.3%) | 1,140 (29.2%) | 569 (10.8%) | 0.1 | 0.05 | 0.41 |

\* Small cell suppressed as per ICES policy.

## Association of patient characteristics with models of care in the last 90 days of life

The mean age of people who received exclusively home-based in-person care was higher (median age 87, IQR, 78–92) than people who received virtual or mixed model care (median

**Table 2. Associations of receiving virtual and mixed model of care in the last 90 days of life according to the type of serious illness using cancer as the main referent group.**

| | Exclusively virtual EOL visits, n (%) | Exclusively home-based in-person visits, n (%) | Mixed EOL visits, n (%) | Exclusively virtual EOL visits vs Mixed EOL visits | |
| --- | --- | --- | --- | --- | --- |
| | | | | Unadjusted OR (95% CI) | Adjusted OR (95% CI) |
| **Cancer** | 33,819 (57.4%) | 1,951 (3.3%) | 23,124 (39.3%) | 1.0 (ref) | 1.0 (ref) |
| **Chronic organ failure** | 11,674 (72.5%) | 495 (3.1%) | 3,927 (24.4%) | 2.03 (1.95, 2.12) | 1.61 (1.54, 1.68) |
| **Dementia** | 2,171 (55.6%) | 255 (6.5%) | 1,482 (37.9%) | 1.00 (0.94, 1.07) | 1.05 (0.98, 1.13) |
| **Multimorbidity** | 3,616 (68.9%) | 162 (3.1%) | 1,468 (28.0%) | 1.68 (1.58, 1.79) | 1.49 (1.39, 1.59) |

*Groups are not mutually exclusive (e.g., individuals with dementia could also be in the chronic organ failure group).

Models were adjusted for age, sex, ethnicity, comorbidities, rurality, neighbourhood income and hospital frailty risk score.

Abbreviation: OR = odds ratio

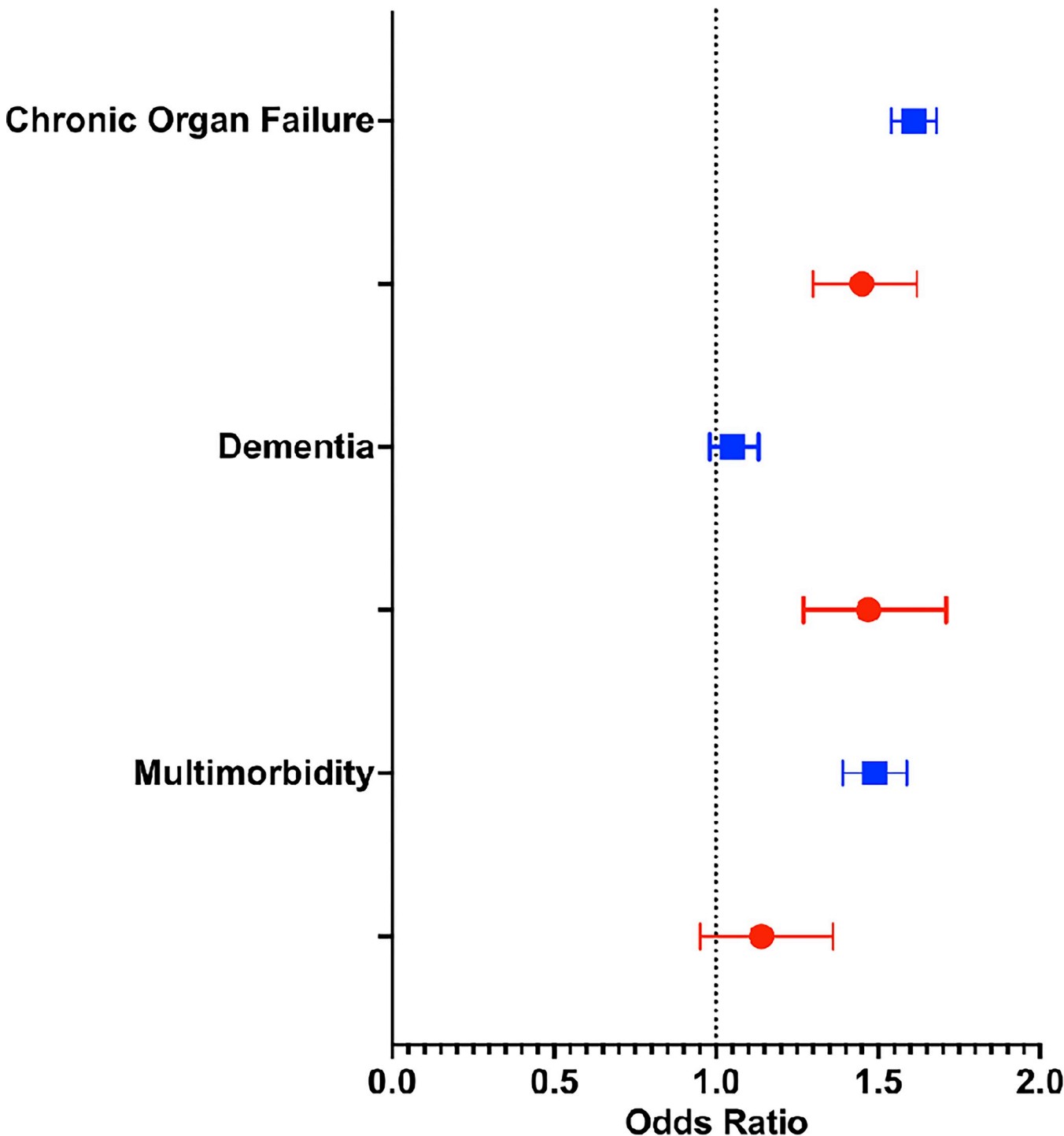

**Fig 2. Primary analysis.** Forest plot of the adjusted odds ratio comparing different physician-delivered models of care in the last 90 days of life among older adults who died with cancer, chronic organ failure, dementia, and multimorbidity in Ontario between March 14, 2020, and January 24, 2022. The comparison includes models of exclusively virtual vs mixed model care (blue) and exclusively home-based in-person vs mixed model care (red) groups. An odds ratio (OR) of >1 implies that the event is more likely to occur in the first group. Models were adjusted for all baseline patient characteristics except the prevalent chronic conditions under study.

age 77, IQR, 68–85 and median age 80, IQR, 70–88 respectively). More females (57%) utilized exclusive home-based in-person care than males (42%). In contrast, males received a higher rate of exclusive virtual care (55%) than females (44%) (S5 Table).

## Discussion

This cohort study of 75,930 people in their last 90 days of life found that a person's type of serious illness (cancer versus terminal noncancer illness) was associated with different physician-delivered care models near the EOL. People with cancer were more likely to receive a mixed model of in-person and virtual care in their last 90 days of life compared to those people with terminal noncancer illnesses. In contrast, people with dementia were more likely to receive exclusively in-person care in their homes, and people with chronic organ failure and multimorbidity were more likely to receive exclusively virtual care compared to people with cancer.

Our study describes differences in patterns of healthcare delivery by physicians near the EOL and identifies differences in care delivery according to a person's type of serious illness. These differences may reflect needs or preferences, differences in physician bias or resources, or abilities to use different modalities of EOL care (i.e., telephone, video, in-person). Although we adjusted our statistical models for multiple potential confounders, the observed differences may also reflect residual confounding due to unmeasured factors such as the effects of the pandemic on access to healthcare and its delivery, or individual patient preferences for specific care modalities. A better understanding of why these differences exists bears further study to ensure people receive optimal care delivery models at the EOL. Furthermore, our findings inform future healthcare resource planning to accommodate different delivery models based on an individual's chronic illness. For example, governments may need to increase and provide additional home care resources to deliver care to patients living with dementia in their homes. These additional supports will require financial investments and policies on supporting reimbursement from payers. In contrast, there may be a need for additional investments in virtual care near the EOL for heart failure patients, ensuring that reimbursement is provided at 100%, among other tailored strategies. This is particularly relevant given the widespread increase in the use of virtual care that occurred during the pandemic with recent modifications to reimbursement policies provided by governments. However, the applicability of these findings to other jurisdictions, such as the United States, may be limited given that a large majority of care near the EOL is delivered by non-physician care providers, often in hospice settings.

Several factors may account for our findings, which are related to those that influence the likelihood of receiving different models of care near the EOL. Resources such as palliative care that are able to support people in their homes (both in person and remotely) may be more readily available and widely used for people with cancer than for people with terminal noncancer illnesses [4, 31]. There may also be underlying differences in the care needs of people with cancer and those with terminal noncancer illnesses, which may influence the choice of care delivery model that a physician uses. For example, people with advanced cancer may have a higher burden and severity of pain crises that require in-person care to be managed effectively. Once addressed, there may be relatively stable periods where symptoms can be managed using exclusively virtual or telephone care, resulting in a mixed model of care. On the other hand, people with heart failure commonly suffer with shortness of breath, which can have a more insidious onset and may, therefore, be managed remotely through intermittent telephone check-ins with resulting adjustments in therapies [32]. People with dementia may require more in-person support and social interaction due to their underlying cognitive decline, functional impairments, and caregiver needs [33].

Our study findings are consistent with prior research demonstrating that a person's type of serious illness, such as cancer, is associated with enhanced access to and use of high-quality models of care near the EOL [4, 5]. Although current evidence is limited, it suggests that a mixed care model may be optimal [16, 30]. A recent scoping review suggested that the benefits of a mixed model of care include increased access to healthcare professionals in a timely manner, improved feelings of safety and security among people, and building genuine relationships between them [9]. Other related research found that healthcare professionals understand the need for virtual care alongside in-person care and emphasized the need for a mixed model approach in specialized community palliative care programs [34]. A recent study proposed that decisions regarding the utilization of virtual or in-person care models near the EOL should be made on a case-by-case basis and according to a person's specific circumstance [35]. Therefore, our study advances the field of knowledge by demonstrating persistent disease-specific differences in care delivery, the causes of which bear further study.

## Strengths and limitations

The study's major strength lies in its population-level coverage. The inclusion of a diverse range of participants from a large population ensures the findings represent the population as a whole, enhancing the study's generalizability and minimizing selection bias. In addition, validated methods used to classify prevalent diseases, including cancer, dementia, and other chronic conditions, minimize misclassification and increase the accuracy and consistency of the findings. Furthermore, we used specific virtual fee codes to identify the models of care for adults in the last 90 days of life. These codes are precise and well-defined, ensuring the outcomes are clearly and objectively measured.

This study also has limitations. First, we studied care delivery during the COVID-19 pandemic period as an opportunity to evaluate these different care models when the use of exclusively virtual care significantly increased. As care delivery was substantially altered during the pandemic, it will be important to evaluate our findings in post-pandemic periods. Second, the administrative databases that we used do not measure a person's individual care needs, preferences, or perceived quality of care, which may direct the model of care they received in their last 90 days of life. Prior complimentary qualitative research conducted by our team through interviews with patients, caregivers, and clinicians identified preferences toward use of a mixed model of care tailored to the preferences and needs of the individual.[16] Third, we measured different models of physician-delivered care in the last 90 days of life but did not measure models of care delivery by other important healthcare providers, including nurse practitioners and social workers. Fourth, people with cancer often have multiple other chronic conditions which may influence their care needs and corresponding care delivery model. Fifth, we did not measure the cause of death, including deaths due to COVID-19 or those exacerbated by COVID-19, as these data were not available.

## Conclusion

A person's type of serious illness was associated with different physician-delivered care models near the EOL. This study demonstrates persistent disease-specific differences in care delivery or possibly the tailoring of models of care in the last 90 days of life based on a person's specific care needs.

## Supporting information

**S1 Table. Reporting of studies Conducted using Observational Routinely collected health Data (RECORD) checklist.**
(DOCX)

**S2 Table. List of virtual care fee codes according to pandemic time periods.**
(DOCX)

**S3 Table. Physician demographics (as of first end-of-life visit for people the physician saw in each disease cohort).**
(DOCX)

**S4 Table. Associations of receiving exclusively home-based in-person and mixed model of care in the last 90 days of life according to the type of serious illness using cancer as main referent group.**
(DOCX)

**S5 Table. Baseline characteristics of the study cohort according to models of care in the last 90 days of life.**
(DOCX)

## Acknowledgments

The authors would like to express their gratitude to our patient and caregiver partner who assisted in the development of the research questions and interpretation of the main findings. They politely declined the offer to include them as co-authors in recognition of their valuable contributions. We thank IQVIA Solutions Canada Inc. for the use of their Drug Information File.

## Author Contributions

**Conceptualization:** Allan S. Detsky, James Downar, Kieran L. Quinn.

**Formal analysis:** Hannah Chung.

**Funding acquisition:** Kieran L. Quinn.

**Supervision:** Kieran L. Quinn.

**Writing – review & editing:** Rabia Akhter, Thérèse A. Stukel, Hannah Chung, Chaim M. Bell, Allan S. Detsky, James Downar, Sarina R. Isenberg, John Lapp, Media Mokhtarnia, Nathan Stall, Peter Tanuseputro, Kieran L. Quinn.

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
