## [Decision Letter · Decision Letter 0]

22 Jan 2024

PONE-D-23-35922Comparison of physician-delivered models of virtual and home-based in-person care for adults in the last 90 days of life with cancer and terminal noncancer illness during the COVID-19 pandemicPLOS ONE

Dear Dr. Quinn,

Thank you for submitting your manuscript to PLOS ONE. After careful consideration, we feel that it has merit but does not fully meet PLOS ONE’s publication criteria as it currently stands. Therefore, we invite you to submit a revised version of the manuscript that addresses the points raised during the review process.

We look forward to receiving your revised manuscript.

Kind regards,

Kartikeya Rajdev, MD

Academic Editor

PLOS ONE

Journal Requirements:

This study received funding from the Canadian Institutes of Health Research (CIHR PNN-177923), and Health Canada’s Health Care Policy and Strategies Program. This study was also supported by the Innovation Fund of the Alternative Funding Plan for the Academic Health Sciences Centres of Ontario as well as ICES, which is funded by an annual grant from the Ontario Ministry of Health and the Ministry of Long-Term Care.  This document used data adapted from the Statistics Canada Postal Code OM Conversion File, which is based on data licensed from Canada Post Corporation, and/or data adapted from the Ontario Ministry of Health Postal Code Conversion File, which contains data copied under license from ©Canada Post Corporation and Statistics Canada. Parts of this material are based on data and/or information compiled and provided by the Ontario Ministry of Health and the Canadian Institute for Health Information. The analyses, conclusions, opinions and statements expressed herein are solely those of the authors and

do not reflect those of the funding or data sources; no endorsement is intended or should be inferred. 

4. In the online submission form, you indicated that The dataset from this study is held securely in coded form at ICES. While data-sharing agreements prohibit ICES from making the dataset publicly available, access may be granted to those who meet pre-specified criteria for confidential access, available at www.ices.on.ca/DAS. The full dataset creation plan and underlying analytic code are available from the authors upon request, understanding that the computer programs may rely upon coding templates or macros that are unique to ICES and are, therefore, either inaccessible or may require modification.

Reviewers' comments:

Reviewer's Responses to Questions

**Comments to the Author**

1. Is the manuscript technically sound, and do the data support the conclusions?

Reviewer #1: Yes

Reviewer #2: Yes

Reviewer #3: Yes

2. Has the statistical analysis been performed appropriately and rigorously? 

Reviewer #1: Yes

Reviewer #2: Yes

Reviewer #3: Yes

3. Have the authors made all data underlying the findings in their manuscript fully available?

Reviewer #1: Yes

Reviewer #2: Yes

Reviewer #3: Yes

4. Is the manuscript presented in an intelligible fashion and written in standard English?

Reviewer #1: Yes

Reviewer #2: Yes

Reviewer #3: Yes

5. Review Comments to the Author

Reviewer #1: Very well designed study

In table 2, I couldnt find data regading in person care, may be i missed it- can you include it.

Was there any change in perception by patients and families regarding quality of care in these models?

Reviewer #2: Several statements were repeated multiple time throughout the article. If possible, to try to minimize the repetitions. Sample size is great. Lot of statistics related calculations had to be used. Appreciate the work from the authors. I liked the tabular format for displaying the statistics. Reducing the repetitions will trim the article size and make it more interesting for readers.

Reviewer #3: Article Review:

Introduction:

The introduction of the article provides a solid foundation but lacks a smooth transition to the study objective. A more seamless connection between the background information and the specific aim would enhance reader comprehension. Additionally, explicitly highlighting the research gap or unknown aspects that the study seeks to address could enhance the introduction's impact, engaging readers and underscoring the study's significance.

Methods:

Compliance with Reporting Guidelines:

The article demonstrates clear reporting in adherence to guidelines. However, a brief explanation of the rationale for choosing linked clinical and administrative datasets would offer readers a deeper understanding of the study design's appropriateness. Including such context could improve the overall transparency of the research.

Exclusion Criteria and Rationale:

The exclusion criteria are well-defined, but the rationale behind each criterion could be more explicitly stated. Providing a clearer justification, such as explaining why individuals with fewer than two visits were excluded, would bolster the methodological rigor and strengthen the study's foundation.

Results:

While the article adjusts for potential confounding variables in the logistic regression models, a brief acknowledgment and discussion of limitations or potential confounders related to these adjustments would enhance the statistical validity of the findings. Addressing this aspect would provide a more comprehensive understanding for readers, ensuring a more robust interpretation of the results.

Discussion:

In the discussion section, there is a mention of implications for healthcare planning, but the article lacks specific recommendations or strategies based on observed differences. Adding actionable insights would enhance the article's relevance for policymakers, ensuring that the research findings are not only informative but also practical in guiding future healthcare planning efforts. Providing concrete suggestions would bridge the gap between research outcomes and practical application, making the discussion more impactful.

6. PLOS authors have the option to publish the peer review history of their article (what does this mean?). If published, this will include your full peer review and any attached files.

Reviewer #1: **Yes: **Naveen Kumar Reddy Tangutur

Reviewer #2: **Yes: **Satyakant Chitturi

Reviewer #3: **Yes: **Vishal Devarkonda

---

## [Author Response · Author response to Decision Letter 0]

21 Feb 2024

Kartikeya Rajdev, MD 

Academic Editor 

PLOS ONE 

PONE-D-23-35922 - "Comparison of physician-delivered models of virtual and home-based in-person care for adults in the last 90 days of life with cancer and terminal noncancer illness during the COVID-19 pandemic”

Thank you for providing us the opportunity to resubmit our work to PLOS One. We are pleased that the editors and reviewers found our study to be important, valuable, relevant, and of considerable interest. Their helpful suggestions and comments have helped us to improve the quality of the manuscript. We have addressed their concerns in detail below.

The following is a summary of the extensive changes we made to the manuscript in response to yours and the reviewer’s suggestions:

1) Provided clear and actionable insights to enhance the study’s relevance for policymakers.

2) Updated Table 2 to report the total number and proportion of people receiving exclusively in-person care.

3) Modified the introduction to improve transition to objective statement.

4) Highlighted the principal knowledge gaps in the introduction.

5) Provided additional rationale in multiple areas in the methods, such as for the exclusion of individuals from the study cohort.

6) Enhanced the interpretation of the main findings through recognition of residual confounding.

7) Updated the limitations to recognize lack of measures on perceptions of quality of care.

8) Reduced repetitive language throughout the manuscript to improve readability and reduce overall length.

EDITOR COMMENTS: 

We updated the manuscript as per PLOS ONE's style requirements.

We added the suggested statement to the online financial disclosure form: “The funders had no role in study design, data collection and analysis, decision to publish, or preparation of the manuscript.” 

3. All PLOS journals now require all data underlying the findings described in their manuscript to be freely available to other researchers, either a. In a public repository, b. Within the manuscript itself, or c. Uploaded as supplementary information. This policy applies to all data except where public deposition would breach compliance with the protocol approved by your research ethics board. If your data cannot be made publicly available for ethical or legal reasons (e.g., public availability would compromise patient privacy), please explain your reasons on resubmission and your exemption request will be escalated for approval. 

We request an exemption from this policy as legal data sharing agreements prohibit ICES from making the dataset publicly available; access may be granted to those who meet pre-specified criteria for confidential access, available at www.ices.on.ca/DAS.

We previously indicated these restrictions in the manuscript as follows,

“The dataset from this study is held securely in coded form at ICES. While data sharing agreements prohibit ICES from making the dataset publicly available, access may be granted to those who meet pre-specified criteria for confidential access, available at www.ices.on.ca/DAS. The full dataset creation plan and underlying analytic code are available from the authors upon request, understanding that the computer programs may rely upon coding templates or macros that are unique to ICES and are therefore either inaccessible or may require modification.” (Data Sharing, page 20, lines 312-317) 

REVIEWERS’ COMMENTS:

Reviewer #1

1. Very well-designed study. In table 2, I couldn’t find data regarding in person care, maybe I missed it - can you include it.

We updated Table 2 to report the total number and proportion of people receiving exclusively in-person care as requested. (Table 2, page 14) We had originally included these data in Supplementary Table 2 because the main comparison of interest was the receipt of exclusively virtual versus mixed models of care.

2. Was there any change in perception by patients and families regarding quality of care in these models? 

We did not measure patient or family perceptions regarding quality of care as these measures are not available in administrative health data in Ontario. We updated the limitations to recognize this important point as follows,

“Second, the administrative databases that we used do not measure a person's individual care needs, preferences, or perceived quality of care, which may direct the model of care they received in their last 90 days of life.” Prior complimentary qualitative research conducted by our team through interviews with patients, caregivers, and clinicians identified preferences toward use of a mixed model of care tailored to the preferences and needs of the individual. (Strengths and Limitations, page 19, lines 287-290)

Reviewer #2

1. Several statements were repeated multiple time throughout the article. If possible, to try to minimize the repetitions. Sample size is great. Lot of statistics related calculations had to be used. Appreciate the work from the authors. I liked the tabular format for displaying the statistics. Reducing the repetitions will trim the article size and make it more interesting for readers. 

Thank you for the suggestion. We reviewed the manuscript and reduced repetitive language throughout to improve readability and reduce overall length.

Reviewer #3

1. Introduction: The introduction of the article provides a solid foundation but lacks a smooth transition to the study objective. A more seamless connection between the background information and the specific aim would enhance reader comprehension. 

We modified the introduction as follows,

“Despite observed differences in EOL care delivery models between people with different types of serious illness before the pandemic, it remains unknown if these differences persisted in the pandemic during a period of widespread virtual care use. Further, it is unknown if a preferential model of mixed model of care is used more frequently among people with cancer than among people with terminal noncancer illness.

To address these existing knowledge gaps in patterns of end-of-life care delivery during the pandemic, we measured the association between types of serious illness (cancer and noncancer terminal illness: chronic organ failure, dementia, multimorbidity) and the use of different physician-delivered care models (exclusively virtual, exclusively home-based in-person, and mixed model of care) near the EOL.” (Introduction, page 3-4, lines 45-53)

2. Introduction: Additionally, explicitly highlighting the research gap or unknown aspects that the study seeks to address could enhance the introduction's impact, engaging readers and underscoring the study's significance. 

We modified the introduction as follows,

“Despite observed differences in EOL care delivery models between people with different types of serious illness before the pandemic, it remains unknown if these differences persisted in the pandemic during a period of widespread virtual care use. Further, it is unknown if a preferential model of mixed model of care is used more frequently among people with cancer than among people with terminal noncancer illness.” (Introduction, page 3-4, lines 45-48)

3. Methods (Compliance with Reporting Guidelines): The article demonstrates clear reporting in adherence to guidelines. However, a brief explanation of the rationale for choosing linked clinical and administrative datasets would offer readers a deeper understanding of the study design's appropriateness. Including such context could improve the overall transparency of the research. 

We added to the methods the following rationale,

“The use of population-level linked administrative data minimizes selection bias and improves the diversity of the study cohort to strengthen the overall generalizability of the study and its principal findings.” (Methods, page 5, lines 65-67)

We previously stated the strengths of using population-level linked clinical and administrative datasets in the discussion as follows,

“The study's major strength lies in its population-level coverage. The inclusion of a diverse range of participants from a large population ensures the findings represent the population as a whole, enhancing the study's generalizability and minimizing selection bias.” (Strengths and Limitations, page 18, lines 276-278) 

4. Methods (Exclusion Criteria and Rationale): The exclusion criteria are well-defined, but the rationale behind each criterion could be more explicitly stated. Providing a clearer justification, such as explaining why individuals with fewer than two visits were excluded, would bolster the methodological rigor, and strengthen the study's foundation. 

We added the following clarification and rationale,

“We excluded adults 1) whose last 90 days of life started before March 14, 2020, which preceded the onset of the pandemic and the majority of resulting health system changes;…” (Methods, page 6, lines 83-84)

“We also excluded individuals who were non-Ontario residents, ineligible for OHIP for more than 90 days continuously in the prior year and therefore could not receive publicly insured health services, and people who did not access the Ontario healthcare system at least once in the past 10 years to ensure people are still residing in the province. (Methods, page 6, lines 90-94)

We provided rationale for the exclusion of individuals with fewer than two visits in our original manuscript as follows,

“The need to have 2 or more visits was necessary to determine if people received exclusively in-person, exclusively virtual, or mixed models of physician-delivered care in the last 90 days of life (the primary outcome).” (Methods, page 6, lines 88-90)

We provided rationale for the remaining exclusion criteria in our original manuscript as follows,

“…2) without prevalent cancer, chronic organ failure, or dementia at 90 days before death (the primary exposures); 3) residents living in nursing homes in the past two years as care delivery during the pandemic was significantly interrupted in this setting; and 4) with ≤1 visit or who were institutionalized during the last 90 days of life. The need to have 2 or more visits was necessary to determine if people received exclusively in-person, exclusively virtual, or mixed models of physician-delivered care in the last 90 days of life (the primary outcome).” (Methods, page 6, lines 84-90) 

5. Results: While the article adjusts for potential confounding variables in the logistic regression models, a brief acknowledgment and discussion of limitations or potential confounders related to these adjustments would enhance the statistical validity of the findings. Addressing this aspect would provide a more comprehensive understanding for readers, ensuring a more robust interpretation of the results. 

We added the following to the interpretation of the principal study findings,

“Although we adjusted our statistical models for multiple potential confounders, the observed differences may also reflect residual confounding due to unmeasured factors such as the effects of the pandemic on access to healthcare and its delivery or individual patient preferences for specific care modalities.” (Discussion, page 16, lines 229-233) 

6. Discussion: In the discussion section, there is a mention of implications for healthcare planning, but the article lacks specific recommendations or strategies based on observed differences. Adding actionable insights would enhance the article's relevance for policymakers, ensuring that the research findings are not only informative but also practical in guiding future healthcare planning efforts. Providing concrete suggestions would bridge the gap between research outcomes and practical application, making the discussion more impactful. 

We modified the discussion of the application of our findings as follows,

“Furthermore, our findings inform future healthcare resource planning to accommodate different delivery models based on an individual's chronic illness. For example, governments may need to increase and provide additional home care resources to deliver care to patients living with dementia in their homes. These additional supports will require financial investments and policies on supporting reimbursement from payers. In contrast, there may be a need for additional investment in virtual care near the EOL for heart failure patients, ensuring that reimbursement is provided at 100%, among other tailored strategies. This is particularly relevant given the widespread increase in use of virtual care that occurred during the pandemic with recent modifications to reimbursement policies provided by governments.” (Discussion, page 16-17, lines 234-243)

---

## [Decision Letter · Decision Letter 1]

25 Mar 2024

Comparison of physician-delivered models of virtual and home-based in-person care for adults in the last 90 days of life with cancer and terminal noncancer illness during the COVID-19 pandemic

PONE-D-23-35922R1

Dear Dr. Quinn,

We’re pleased to inform you that your manuscript has been judged scientifically suitable for publication and will be formally accepted for publication once it meets all outstanding technical requirements.

Kind regards,

Kartikeya Rajdev, MD

Academic Editor

PLOS ONE

Additional Editor Comments (optional):

Reviewers' comments:

Reviewer's Responses to Questions

**Comments to the Author**

1. If the authors have adequately addressed your comments raised in a previous round of review and you feel that this manuscript is now acceptable for publication, you may indicate that here to bypass the “Comments to the Author” section, enter your conflict of interest statement in the “Confidential to Editor” section, and submit your "Accept" recommendation.

Reviewer #1: All comments have been addressed

Reviewer #2: All comments have been addressed

Reviewer #3: All comments have been addressed

2. Is the manuscript technically sound, and do the data support the conclusions?

Reviewer #1: Yes

Reviewer #2: Yes

Reviewer #3: Yes

3. Has the statistical analysis been performed appropriately and rigorously? 

Reviewer #1: Yes

Reviewer #2: Yes

Reviewer #3: Yes

4. Have the authors made all data underlying the findings in their manuscript fully available?

Reviewer #1: Yes

Reviewer #2: Yes

Reviewer #3: Yes

5. Is the manuscript presented in an intelligible fashion and written in standard English?

Reviewer #1: Yes

Reviewer #2: Yes

Reviewer #3: Yes

6. Review Comments to the Author

Reviewer #1: It was a well designed study. You have answered mine and other reviewer's questions. Agree that we need more prospective studies to get better understanding of different models of care that can help patients during end of life.

Reviewer #2: Thank you for addressing our concerns and modifying the language in the article. Would be ideal to publish this article specifically in Covid section. Length of the article is good now. Tabular format for statistics is easy for review.

Reviewer #3: Great work! The manuscript offers a thorough examination of the correlation between various serious illnesses and the utilization of physician-delivered care models near the end-of-life (EOL) during the pandemic. This population-based cohort study, conducted in Ontario, Canada, showcases meticulous attention to detail in both design and methodology. Leveraging health administrative datasets, the authors amassed a substantial sample of 75,930 adults aged 18 years and older in their final 90 days of life, encompassing individuals who passed away from cancer or terminal noncancer conditions. This ample sample size bolsters the study's credibility and widens its applicability.

The findings of the study provide fascinating insights into the utilization patterns of different physician-delivered care models among individuals with cancer and noncancer illnesses. Notably, a larger proportion of cancer patients received a mixed model of care compared to those with noncancer conditions. Furthermore, the analysis unveils disease-specific variations in care delivery, with individuals afflicted with chronic organ failure and multimorbidity exhibiting greater odds of receiving virtual care compared to cancer patients. Likewise, individuals with dementia were more inclined to receive home-based in-person care as opposed to other care models.

The implications drawn from this study are profound, emphasizing the necessity of tailoring care models to suit the unique needs of individuals with diverse serious illnesses nearing the EOL. By pinpointing enduring disease-specific disparities in care provision, the study underscores the imperative of embracing patient-centered approaches to end-of-life care. In sum, the manuscript furnishes invaluable insights to the existing body of literature and lays the groundwork for future research endeavors in this pivotal realm of healthcare delivery.

7. PLOS authors have the option to publish the peer review history of their article (what does this mean?). If published, this will include your full peer review and any attached files.

Reviewer #1: **Yes: **Naveen Kumar Reddy Tangutur

Reviewer #2: No

Reviewer #3: **Yes: **Vishal Devarkonda

---

## [Editor Report · Acceptance letter]

9 Aug 2024

PONE-D-23-35922R1 

PLOS ONE

Dear Dr. Quinn, 

I'm pleased to inform you that your manuscript has been deemed suitable for publication in PLOS ONE. Congratulations! Your manuscript is now being handed over to our production team.

Kind regards, 

on behalf of

Dr. Kartikeya Rajdev 

Academic Editor

PLOS ONE